# AViD Dataset: Anonymized Videos from Diverse Countries

**AJ Piergiovanni**
Indiana University
ajpiergi@indiana.edu

**Michael S. Ryoo**
Stony Brook University
mryoo@cs.stonybrook.edu

## Abstract

We introduce a new public video dataset for action recognition: Anonymized Videos from Diverse countries (AViD). Unlike existing public video datasets, AViD is a collection of action videos from many different countries. The motivation is to create a public dataset that would benefit training and pretraining of action recognition models for everybody, rather than making it useful for limited countries. Further, all the face identities in the AViD videos are properly anonymized to protect their privacy. It also is a static dataset where each video is licensed with the creative commons license. We confirm that most of the existing video datasets are statistically biased to only capture action videos from a limited number of countries. We experimentally illustrate that models trained with such biased datasets do not transfer perfectly to action videos from the other countries, and show that AViD addresses such problem. We also confirm that the new AViD dataset could serve as a good dataset for pretraining the models, performing comparably or better than prior datasets[1].

## 1 Introduction

Video recognition is an important problem with many potential applications. One key challenge in training a video model (e.g., 3D spatio-temporal convolutional neural networks) is the lack of data, as these models generally have more parameters than image models requiring even more data. Kinetics (Kay et al., 2017) found that by training on a hundreds of thousands of labeled video clips, one is able to increase the performance of video models significantly. Other large-scale datasets, such as HVU (Diba et al., 2019), Moments-in-Time (Monfort et al., 2018), and HACS (Zhao et al., 2019) also have been introduced, motivated by such findings.

However, many of today's large-scale datasets suffer from multiple problems: First, due to their collection process, the videos in the datasets are very biased particularly in terms of where the videos are from (Fig. 1 and Table 3). Secondly, many of these datasets become inconsistent as YouTube videos get deleted. For instance, in the years since Kinetics-400 was first released, over 10% of the videos have been removed from YouTube. Further, depending on geographic location, some videos may not be available. This makes it very challenging for researchers in different countries and at different times to equally benefit from the data and reproduce the results, making the trained models to be biased based on when and where they were trained. They are not static datasets (Figure 3).

AViD, unlike previous datasets, contains videos from diverse groups of people all over the world. Existing datasets, such as Kinetics, have videos mostly from from North America (Kay et al., 2017) due to being sampled from YouTube and English queries. AViD videos are distributed more broadly across the globe (Fig. 1) since they are sampled from many sites using many different languages. This is important as certain actions are done differently in different cultures, such as greetings (shown

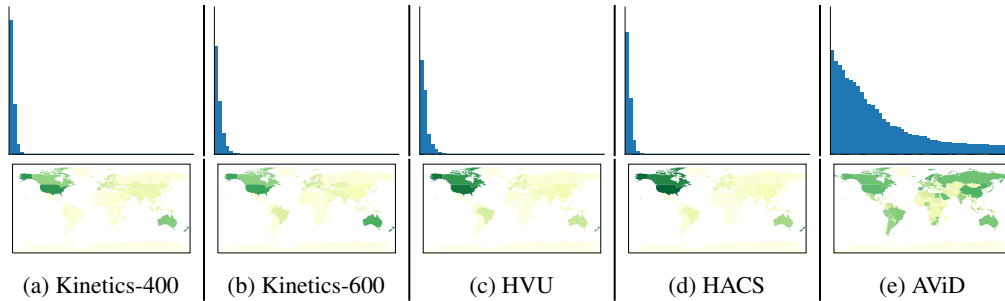

| (a) Kinetics-400 | (b) Kinetics-600 | (c) HVU | (d) HACS | (e) AViD |

Figure 1: Histogram and Heatmap describing geological distributions of videos for Kinetics and AViD. Video locations are obtained from their geotags using the public YouTube API (check Appendix for details). X-axis of the above histogram correspond to different countries and Y-axis correspond to the number of videos. The color in heatmap is proportional to the number of videos from each country. Darker color means more videos. As shown, AViD has more diverse videos than the others.

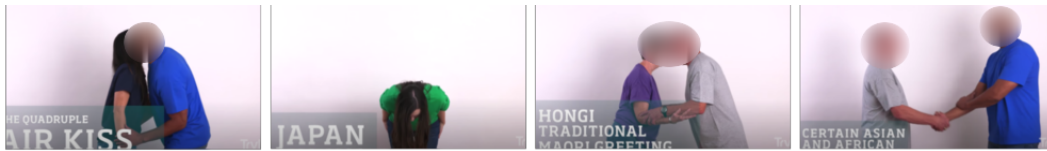

Figure 2: Examples of 'greeting' in four different countries. Without diverse videos from all over the world, many of these would not be labeled as 'greeting' by a model. These examples are actual video frames from the AViD dataset.

in Fig. 2), nodding, etc. As many videos contain text, such as news broadcasts, the lack of diversity can further bias results to rely on English text which may not be present in videos from different regions of the world. Experimentally, we show diversity and lack of diversity affects the recognition.

Further, we anonymize the videos by blurring all the faces. This prevents humans and machines from identifying people in the videos. This is an important property for institutions, research labs, and companies respecting privacy to take advantage the dataset. Due to this fact, face-based actions (e.g., smile, makeup, brush teeth, etc.) have to be removed as they would be very difficult to recognize with blurring, but we show that the other actions are still reliably recognized.

Another technical limitation with YouTube-based datasets including Kinetics, ActivityNet (Caba Heilbron et al., 2015), YouTube-8M (Abu-El-Haija et al., 2016), HowTo100M (Miech et al., 2019), AVA (Gu et al., 2017) and others, is that downloading videos from YouTube is often blocked. The standard tools for downloading videos can run into request errors (many issues on GitHub exist, with no permanent solution). These factors limit many researchers from being able to use large-scale video datasets.

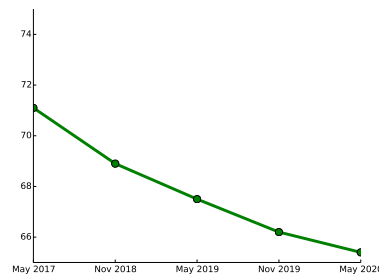

Figure 3: Performance of Kinetics-400 over time as more videos are removed from YouTube. The performance is constantly dropping.

To address these challenges, we introduce a new, large-scale dataset designed to solve these problems. The key benefits of this dataset is that it captures the same actions as Kinetics plus hundreds of new ones. Further, we choose videos from a variety of sources (Flickr, Instagram, etc.) that have a creative-commons licence. This license allows us to download, modify and distribute the videos as needed. We create a **static** video dataset that can easily be downloaded. We further provide tags based on the user-generated tags for the video, enabling studying of weakly-labeled data learning. Also unique is the ability to add 'no action' which we show helps in action localization tasks. To summarize,

- AViD contains actions from diverse countries obtained by querying with many languages.
- AViD is a dataset with face identities removed
- AViD is a static dataset with all the videos having the creative-commons licence.

## 2 Dataset Creation

The dataset creation process follows multiple steps. First we generated a set of action classes. Next, we sampled videos from a variety of sources to obtain a diverse sample of all actions. Then we generate candidate clips from each video. These clips are then annotated by human. We now provide more details about this process.

### 2.1 Action Classes

Unlike images, where objects are clearly defined and have physical boundaries, determining an action is in videos is a far more ambiguous task. In AViD, we follow many previous works such as Kinetics (Kay et al., 2017), where an action consists of a verb and a noun when needed. For example, 'cutting apples' is an action with both a verb and noun while 'digging' is just verb.

To create the AViD datasets, the action classes begin by combining the actions in Kinetics, Charades, and Moments in Time, as these cover a wide variety of possible actions. We then remove all actions involving the face (e.g., 'smiling,' 'eyeliner,' etc.) since we are blurring faces, as this makes it extremely difficult to recognize these actions. Note that we do leave actions like 'burping' or 'eating' which can be recognized by other contextual cues and motion. We then manually combine duplicate/similar actions. This resulted in a set of 736 actions. During the manual annotation process, we allowed users to provide a text description of the actions in the video if none of the candidate actions were suitable and the additional 'no action' if there was no action in the video. Based on this process, we found another 159 actions, resulting in 887 total actions. Examples of some of the new ones are 'medical procedures,' 'gardening,' 'gokarting,' etc.

Previous works have studied using different forms of actions, some finding actions associated with nouns to be better (Sigurdsson et al., 2017) while others prefer atomic, generic action (Gu et al., 2017). The Moments in Time (Monfort et al., 2018) takes the most common verbs to use as actions, while Charades (Sigurdsson et al., 2016) uses a verb and noun to describe each action. Our choice of action closely follows these, and we further build a hierarchy that will enable studying of verb-only actions compared to verb+noun actions and levels of fine-grained recognition.

#### 2.1.1 Hierarchy

After deciding the action classes, we realized there was a noticeable hierarchy capturing these different actions. Hierarchies have been created for ImageNet (Deng et al., 2009) to represent relationships such as fine-grained image classification, but they have not been widely used in video understanding. ActivityNet (Caba Heilbron et al., 2015) has a hierarchy, but is a smaller dataset and the hierarchy mostly capture broad differences and only has 200 action classes.

We introduce a hierarchy that captures more interesting relationships between actions, such as 'fishing' → 'fly tying,' 'casting fishing line,' 'catching fish,' etc. And more broad differences such as 'ice fishing' and 'recreational fishing.' Similarly, in the 'cooking class' we have 'cutting fruit' which has both 'cutting apples' and 'cutting pineapple'. Some actions, like 'cutting strawberries' didn't provide enough clips (e.g., less than 10), and in such case, we did not create the action category and made the videos only belong to the 'cutting fruit' class. This hierarchy provides a starting point to study various aspects of what an action is, and how we should define actions and use the hierarchy in classifiers. Part of the hierarchy is shown in Fig. 4, the full hierarchy is provided in the supplementary material.

### 2.2 Video Collection

AViD videos are collected from several websites: Flickr, Instagram, etc. But we ensure all videos are licensed with the creative commons license. This allows us to download, modify (blur faces), and distribute the videos. This enables the construction of a static, anonymized, easily downloadable video dataset for reproducible research.

In order to collect a *diverse* set of candidate videos to have in the dataset, we translated the initial action categories into 22 different languages (e.g., English, Spanish, Portuguese, Chinese, Japanese, Afrikaans, Swahili, Hindi, etc.) covering every continent. We then searched multiple video websites (Instagram, Flickr, Youku, etc.) for these actions to obtain initial video samples. This process resulted

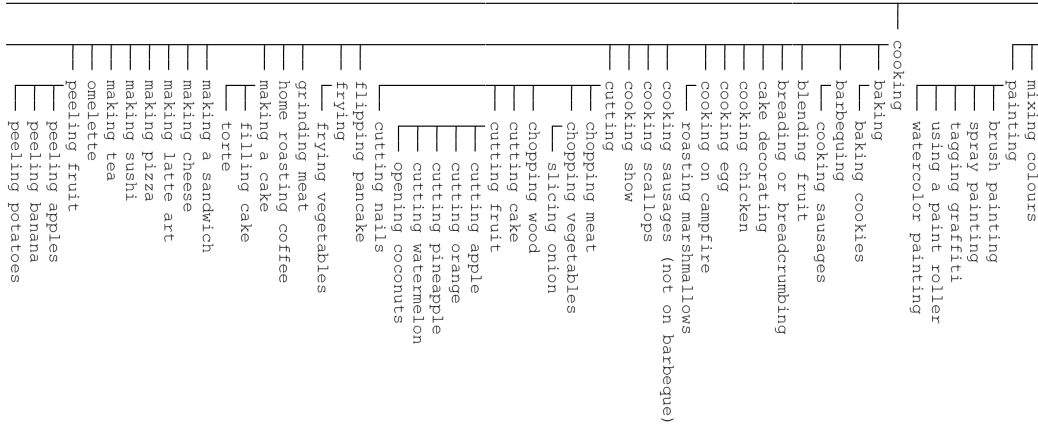

Figure 4: Illustration of a section of the hierarchy of activities in AViD. Check Appendix for the full hierarchy with 887 classes.

Table 1: Comparison of large video datasets for action classification.

| Dataset | Classes | Train Clips | Test Clips | Hours | Clip Dur. |
|---|---|---|---|---|---|
| Kinetics-400 | 400 | 230k | 20k | 695 | 10s |
| Kinetics-600 | 600 | 392k | 30k | 1172 | 10s |
| Moments in Time | 339 | 802k | 33k | 667 | 3s |
| AViD | 887 | 410k | 40k | 880 | 3-15s |

in a set of 800k videos. From these videos, we took multiple sample clips. As shown in Fig. 1, this process found videos from all over the globe.

We ensured there was no overlap of AViD videos and those in the validation or testing sets of Kinetics. There is some minor overlap between some of AViD videos and the training set of Kinetics, which is an outcome due to that the both datasets were collected from the web.

## 2.3 Action Annotation

We annotate the candidate clips using Amazon Mechanical Turk. In order to make human annotations more efficient, we use I3D model (Carreira and Zisserman, 2017) to generate a set of potential candidate labels for each clip (the exact number depends on how many actions I3D predicted, usually 2-3) and provide them as suggestions to the human annotators. We also provide annotators an option to select the 'other' and 'none' category and manually specify what the action is. For each task, one of the videos was from an existing dataset where the label was known. This served as a quality check and the annotations were rejected if the worker did not correctly annotate the test video. A subset of the videos where I3D (trained with Kinetics) had very high confidence (> 90%) were verified manually by the authors.

As a result, a total of 500k video clips were annotated. Human annotators labeled 300k videos manually, and 200k videos with very high-confidence I3D predictions were checked by the authors and the turkers. Of these, about 100k videos were labeled as the 'other' action by the human annotators, suggesting that I3D + Kinetics training does not perform well on these actions. Of these, about 50k videos were discarded due to poor labeling or other errors, resulting in a dataset of 450k total samples.

We found the distribution of actions follows a Zipf distribution (shown in Fig. 5, similar to the observation of AVA (Gu et al., 2017). We split the dataset into train/test sets by taking 10% of each class as the test videos. This preserves the Zipf distribution.

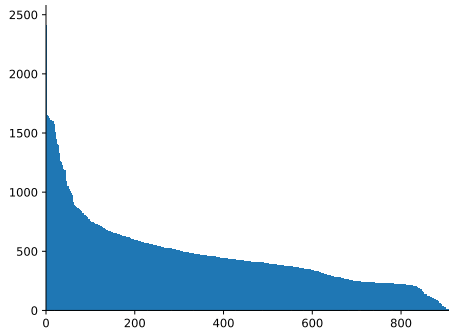

Figure 5: Distribution of videos per class in the AViD dataset. We find it follows a Zipf distribution, similar to the actions in other large-scale video datasets.

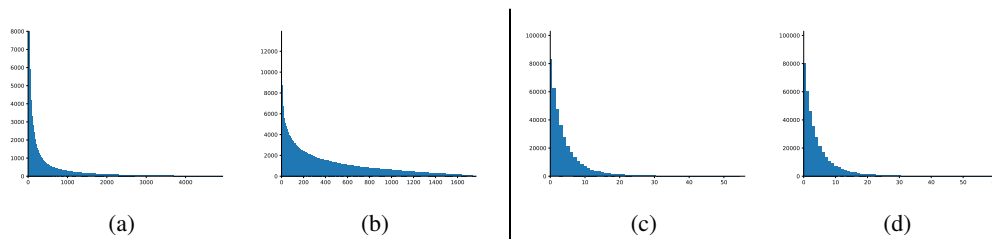

| (a) | (b) | (c) | (d) |

Figure 6: Evaluation of the weak tag distributions. **(a/b)** Number of times each tag appears in the dataset from the agglomerative clustering or affinity propagation. **(c/d)** Number of tags in each video. Videos have between 0 and 65 tags, most have 1-8 tags.

## 2.4 Weak Tag Annotation

In addition to action category annotation per video clips, AviD dataset also provides a set of weak text tags. To generate the weak tags for the videos, we start by translating each tag (provided from the web) into English. We then remove stopwords (e.g., 'to,' 'the,' 'and,' etc.) and lemmatize the words (e.g., 'stopping' to 'stop'). This transforms each tag into its base English word.

Next, we use word2vec (Mikolov et al., 2013) to compute the distance between each pair of tags, and use affinity propagation and agglomerative clustering to generate 1768 and 4939 clusters, respectively. Each video is then tagged based on these clusters. This results in two different sets of tags for the videos, both of which are provided for further analysis, since it is unclear which tagging strategy will more benefit future approaches. The overall distribution of tags is shown in Fig. 6, also following an exponential distribution.

## 3 Experiments

We conducted a series of experiments with the new AViD dataset. This not only includes testing existing video CNN models on the AViD dataset and further evaluating effectiveness of the dataset for pretraining, but also includes quantitative analysis comparing different datasets. Specifically, we measure video source statistics to check dataset biases, and experimentally confirm how well a model trained with action videos from biased countries generalize to videos from different countries. We also evaluate how face blurring influences the classification accuracy, and introduce weak annotations of the dataset.

**Implementation Details** We implemented the models in PyTorch and trained them using four Titan V GPUs. To enable faster learning, we followed the multi-grid training schedule (Wu et al., 2019). The models, I3D (Carreira and Zisserman, 2017), 2D/(2+1D)/3D ResNets (He et al., 2016; Tran et al., 2018, 2014), Two-stream (Simonyan and Zisserman, 2014), and SlowFast (Feichtenhofer et al.,

Table 2: Performance of multiple baselines models on the AViD dataset.

| Model | Acc (conv) | Acc (multi-crop) |
|---|---|---|
| 2D ResNet-50 | 36.2% | 35.3% |
| I3D (Carreira and Zisserman, 2017) | 46.5% | 46.8% |
| 3D ResNet-50 | 47.9% | 48.2% |
| Two-Stream 3D ResNet-50 | 49.9% | 50.1% |
| Rep-Flow ResNet-50 (Piergiovanni and Ryoo, 2019a) | 50.1% | 50.5% |
| (2+1)D ResNet-50 | 46.7% | 48.8% |
| SlowFast-50 4x4 (Feichtenhofer et al., 2018) | 48.5% | 47.4% |
| SlowFast-50 8x8 (Feichtenhofer et al., 2018) | 50.2% | 50.4% |
| SlowFast-101 16x8 (Feichtenhofer et al., 2018) | 50.8% | 50.9% |

2018), were trained for 256 epochs. The learning rate followed a cosine decay schedule with a max of 0.1 and a linear warm-up for the first 2k steps. Each GPU used a base batch size of 8 clips, which was then scaled according to the multi-grid schedule (code provided in supplementary materials). The base clip size was 32 frames at $224 \times 224$ image resolution.

For evaluation, we compared both convolutional evaluation where the entire $T$ frames at $256 \times 256$ were given as input as well as a multi-crop evaluation where 30 random crops of 32 frames at $224 \times 224$ are used and the prediction is the average over all clips.

**Baseline Results**  In Table 2, we report the results of multiple common video model baseline networks. Overall, our findings are consistent with the literature.

**Diversity Analysis**  Since AViD is designed to capture various actions from diverse countries, we conduct a set of experiments to measure the diversity and determine the effect of having diverse videos.

First, we computed geo-location statistics of AViD and other datasets, and compared them. To obtain the locations of AViD videos, we extract the geo-tagged location for videos where it was available (about 75% of total AViD videos). We used the public API of the site where each AViD video came from to gather the geolocation statistics. Similarly, we used the public YouTube API to gather the geolocation statistics for the Kinetics, HACS, and HVU videos. Further, after the initial release of AViD (on arXiv), the Kinetics team provided us their location statistics estimate (Smaira et al., 2020). As it is a bit different from our estimate, we also directly include such data for the comparison.[2]

To measure the diversity of each dataset, we report a few metrics: (1) percentage of videos in North America, Latin America, Europe, Asia, and Africa. (2) As a proxy for diversity and bias, we assume a uniform distribution over all countries would be the most fair (this assumption is debatable), then using the Wasserstein distance, we report the distance from the distribution of videos to the uniform distribution. The results are shown in Table 3. We note that due to the large overlap in videos between HVU and Kinetics-600, their diversity stats are nearly identical. Similarly, as HACS is based on English queries of YouTube, it also results in a highly North American biases dataset. We note that Kinetics-600 and -700 made efforts to improve diversity by querying in Spanish and Portuguese, which did improve diversity in those countries (Carreira et al., 2018; Smaira et al., 2020).

In addition, we ran an experiment training the baseline model on each dataset, and testing it on videos from different regions of the world. Specifically, we train the baseline 3D ResNet model with either Kinetics-400/600 or AViD. Then we evaluated the models on AViD videos using action classes shared by both Kinetics-400 and AViD (about 397 classes) while splitting evaluation into North American, Rest of World, or other regions. The results are summarized in Table 4. We find that the models trained with any of the three datasets perform quite similarly on the North American videos. However, the Kinetics trained models do not perform as well on the diverse videos, while AViD models show a much smaller drop. This suggests that current datasets do not generalize well to diverse world data, showing the importance of building diverse datasets. In Table 5, we show the results when using all

Table 3: Comparing diversity of videos based on geotagged data. The table shows percentages of the videos from North America, Latin American, Europe, Asia, and Africa. 'Div' measures the Wasserstein distance between the actual data distribution and the uniform distribution, the lower the more balanced videos are (i.e., no location bias). For Kinetics, we include both our estimated numbers ([†]) as well as the internal numbers from the Kinetics team (Smaira et al., 2020)[2].

| Dataset | N.A. | L.A. | EU | Asia | AF | Div |
|---|---|---|---|---|---|---|
| Kinetics-400[†] | 96.2 | 0.3 | 2.3 | 1.1 | 0.1 | 0.284 |
| Kinetics-400[2] | 59.0 | 3.4 | 21.4 | 11.8 | 0.8 | 0.169 |
| Kinetics-600[†] | 87.3 | 6.1 | 4.3 | 2.2 | 0.1 | 0.269 |
| Kinetics-600[2] | 59.1 | 5.7 | 19.3 | 11.3 | 0.9 | 0.164 |
| Kinetics-700[2] | 56.8 | 7.6 | 19.6 | 11.5 | 1.0 | 0.158 |
| HVU | 86.4 | 6.3 | 4.7 | 2.5 | 0.1 | 0.266 |
| HACS | 91.4 | 1.5 | 5.8 | 1.2 | 0.1 | 0.286 |
| AViD | 32.5 | 18.6 | 19.7 | 20.5 | 8.7 | 0.052 |

Table 4: Effect of having diverse videos during training. Note that we only test on AViD videos with activities shared between Kinetics-400 and AViD (397 classes). We report the accuracy on North American (N.A.) videos and the rest of the world (RoW) videos, and specific region videos.

| Model | Training Data | Acc (N.A.) | Acc (RoW) | L.A. | EU | Asia | AF |
|---|---|---|---|---|---|---|---|
| 3D ResNet-50 | Kin-400 | 72.8% | 64.5% | 68.3% | 71.2% | 61.5% | 58.4% |
| 3D ResNet-50 | Kin-600 | 73.5% | 65.5% | 69.3% | 72.4% | 62.4% | 59.4% |
| 3D ResNet-50 | AViD (all) | 75.2% | 73.5% | 74.5% | 74.3% | 74.9% | 71.4% |

AViD classes, but using training on a specific region then testing on that region vs. all other regions[3]. We observe that the performance drops when training vs. testing are from different regions. This further suggests that having a training set of videos from diverse countries are essential.

**Fine-tuning** We pretrain several of the models with AViD dataset, and fine-tune on HMDB-51 (Kuehne et al., 2011) and Charades (Sigurdsson et al., 2016).

The objective is to compare AViD with exising datasets in terms of pretraining, including Kinetics-400/600 (Kay et al., 2017) and Moments-in-time (MiT) (Monfort et al., 2018). Note that these results are based on using RGB-only as input; no optical flow is used.

In Table 6, we compare the results on HMDB. We find that AViD performs quite similarly to both Kinetics and MiT. Note that the original Kinetics has far more videos than are currently available (as shown in Figure 3), thus the original fine-tuning performance is higher (indicated in parenthesis).

In Table 7, we compare the results on the Charades dataset. Because the AViD dataset also provides videos with 'no action' in contrast to MiT and Kinetics which only have action videos, we compare the effect of using 'no action' as well. While AViD nearly matches or improves performance even without 'no action' videos in the classification setting, we find that the inclusion of the 'no action'

Table 5: Training on one region and testing on the same and on the others all AViD classes. In all cases, the models perform worse on other regions than the one trained on[3]. This table uses a 3D ResNet-50.

| AViD Training Data | Acc (Same Region) | Acc (All Other Regions) |
|---|---|---|
| N.A. | 51.8% | 42.5% |
| L.A. | 49.4% | 38.5% |
| EU | 47.5% | 39.4% |
| Asia | 46.7% | 41.2% |
| Africa[3] | 42.5% | 32.2% |

Table 6: Performance standard models fine-tuned on HMDB. Numbers in parenthesis are based on original, full Kinetics dataset which is no longer available.

| Model | Pretrain Data | Acc |
|---|---|---|
| I3D (Carreira and Zisserman, 2017) | Kin-400 | 72.5 (74.3) |
| I3D (Carreira and Zisserman, 2017) | Kin-600 | 73.8 (75.4) |
| I3D (Carreira and Zisserman, 2017) | MiT | 74.7 |
| I3D (Carreira and Zisserman, 2017) | AViD | 75.2 |
| 3D ResNet-50 | Kin-400 | 75.7 (76.7) |
| 3D ResNet-50 | Kin-600 | 76.2 (77.2) |
| 3D ResNet-50 | MiT | 75.4 |
| 3D ResNet-50 | AViD | 77.3 |

Table 7: Fine-tuning on Charades using the currently available Kinetics videos. We report results for both classification and the localization setting. We also compare the use of the 'none' action in AViD. [1] (Piergiovanni and Ryoo, 2018)

| Model | Pretrain Data | Class mAP | Loc mAP |
|---|---|---|---|
| I3D (Carreira and Zisserman, 2017) | Kin-400 | 34.3 | 17.9 |
| I3D (Carreira and Zisserman, 2017) | Kin-600 | 36.5 | 18.4 |
| I3D (Carreira and Zisserman, 2017) | MiT | 33.5 | 15.4 |
| I3D (Carreira and Zisserman, 2017) | AViD (- no action) | 36.2 | 17.3 |
| I3D (Carreira and Zisserman, 2017) | AViD | 36.7 | 19.7 |
| 3D ResNet-50 | Kin-400 | 39.2 | 18.6 |
| 3D ResNet-50 | Kin-600 | 41.5 | 19.2 |
| 3D ResNet-50 | MiT | 35.4 | 16.4 |
| 3D ResNet-50 | AViD (- no action) | 41.2 | 18.7 |
| 3D ResNet-50 | AViD | 41.7 | 23.2 |
| 3D ResNet-50 + super-events [1] | AViD | 42.4 | 25.2 |

greatly benefits the localization setting, establishing a new state-of-the-art for Charades-localization (25.2 vs. 22.3 in (Piergiovanni and Ryoo, 2019b)).

**Learning from Weak Tags**   We compare the effect of using the weak tags generated for the AViD dataset compared to using the manually labeled data. The results are shown in Table 8. Surprisingly, we find that using the weak tags provides strong initial features that can be fine-tuned on HMDB without much different in performance. Future works can explore how to best use the weak tag data.

**Blurred Face Effect**   During preprocessing, we use a face detector to blur any found faces in the videos. We utilize a strong Gaussian blur with random parameters. Gaussian blurring can be reversed if the location and parameters are known, however, due to the randomization of the parameters, it would be practically impossible to reverse the blur and recover true identity.

Since we are modifying the videos by blurring faces, we conducted experiments to see how face blurring impacts performance. We compare performance on AViD (accuracy) as well as fine-tuning on HMDB (accuracy) and Charades (mAP) classification. The results are shown in Table 9. While

Table 8: Performance of 3D ResNet-50 using fully-labeled data vs. the weak tags data evaluated on HMDB. 'Aff' is affinity propagation and 'Agg' agglomerative clustering.

| Model | Pretrain Data | Acc |
|---|---|---|
| 3D ResNet-50 | Kin-400 | 76.7 |
| 3D ResNet-50 | AViD | 77.3 |
| 3D ResNet-50 | AViD-weak (Agg) | 76.4 |
| 3D ResNet-50 | AViD-weak (Aff) | 75.3 |

Table 9: Measuring the effects of face blurring on AViD, HMDB and Charades classification. Note that only the faces in AViD are blurred.

| Model | Data | AViD | HMDB | Charades |
|---|---|---|---|---|
| 3D ResNet-50 | AViD-no blur | 48.2 | 77.5 | 42.1 |
| 3D ResNet-50 | AViD-blur | 47.9 | 77.3 | 41.7 |

Table 10: Effect of temporal information in AViD.

| Model | # Frames | In Order | Shuffled |
|---|---|---|---|
| 2D ResNet-50 | 1 | 32.5 | 32.5 |
| 3D ResNet-50 | 1 | 32.5 | 32.5 |
| 3D ResNet-50 | 16 | 44.5 | 38.7 |
| 3D ResNet-50 | 32 | 47.9 | 36.5 |
| 3D ResNet-50 | 64 | 48.2 | 35.6 |

face blurring slightly reduces performance, the impact is not that great. This suggests it has a good balance of anonymization, yet still recognizable actions.

**Importance of Time**   In videos, the use of temporal information is often important when recognizing actions by using optical flow (Simonyan and Zisserman, 2014), stacking frames, RNNs (Ng et al., 2015), temporal pooling (Piergiovanni et al., 2017), and other approaches. In order to determine how much temporal information AViD needs, we compared single-frame models to multi-frame. We then shuffled the frames to measure the performance drop. The results are shown in Table 10. We find that adding more frames benefits performance, while shuffling them harms multi-frame model performance. This suggests that temporal information is quite useful for recognizing actions in AViD, making it an appropriate dataset for developing spatio-temporal video models.

## 4    Conclusions

We present AViD, a new, static, diverse and anonymized video dataset. We showed the importance of collecting and learning from diverse videos, which is not captured in existing video datasets. Further, AViD is **static** and easily distributed, enabling reproducible research. Finally, we showed that AViD produces similar or better results on datasets like HMDB and Charades.

## Broader Impacts

We quantitatively confirmed that existing video datasets for action recognition are highly biased. In order to make people and researchers in diverse countries more fairly benefit from a public action recognition dataset, we propose the AViD dataset. We took care to query multiple websites from many countries in many languages to build a dataset that represents as many countries as possible. We experimentally showed that by doing this, we can reduce the bias of learned models. We are not aware of any other large-scales datasets (with hundreds of video hours) which took such country diversity into the consideration during the collection process.

As this dataset contains a wide variety of actions, it could enable malicious parties to build systems to monitor people. However, we took many steps to preserve the identity of people and eliminate the ability to learn face-based actions, which greatly reduces the negative uses of the data. The positive impacts of this dataset are enabling reproducible research on video understanding which will help more advance video understanding research with consistent and reliable baselines. We emphasize once more that our dataset is a static dataset respecting the licences of all its videos.

## Acknowledgement

This work was supported in part by the National Science Foundation (IIS-1812943 and CNS1814985).

## Footnotes

[1]The dataset is available https://github.com/piergiaj/AViD

[2]We believe the main difference comes from the use of public YouTube API vs. YouTube's internal geolocation metadata estimated based on various factors. Please see the appendix for more details.

[3]There are only ~35k training clips from Africa, and the smaller training set reduces overall performance.

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
