[Supplementary Material]

# –Supplementary Material–
# AViD Dataset: Anonymized Videos from Diverse Countries

**AJ Piergiovanni**
Indiana University
ajpiergi@indiana.edu

**Michael S. Ryoo**
Stony Brook University
mryoo@cs.stonybrook.edu

## A   Diversity Statistics Collection

In order to find the country location for each video in previous YouTube-based datasets (e.g., Kinetics, HACS, etc.), we used the public YouTube API. Specifically, using https://developers.google.com/youtube/v3/docs/videos, we extracted the 'recordingDetails.location' object. Importantly, it notes that

> 'The geolocation information associated with the video. Note that the child property values identify the location that the video owner wants to associate with the video. The value is editable, searchable on public videos, and might be displayed to users for public videos.'

This is the only location data YouTube publicly provides and many videos in existing datasets do not have this field. In our measure, roughly 8% of the videos had such geolocation. We then used reverse-geocode library https://pypi.org/project/reverse-geocode/ to map the coordinates to the country, then manually mapped the countries to each region.

For full transparency, we provide detailed breakdowns of the diversity data we were able to measure with these tools in Table 1 as an example.

| Country | Video Count |
|---|---|
| North America | 32,767 |
| EU | 1,613 |
| Latin America | 2,289 |
| Asia | 938 |
| Africa | 37 |
| No Location | 422,645 |

Table 1: Kinetics-400 Video Distribution

## B   Difference to Kinetics Numbers

After the initial version of AViD was released (on arXiv), the Kinetics team provided numbers based on the estimated upload location of the video (this metadata is not publicly available) (Smaira et al., 2020).

In the paper, we have included their diversity statistics as well, as they are more complete, representing 90% of videos, compared to about 8% that we were able to get geolocation for.

# References

L. Smaira, J. Carreira, E. Noland, E. Clancy, A. Wu, and A. Zisserman. A short note on the kinetics-700-2020 human action dataset, 2020.

# C   Action Classes

1. abseiling
2. acoustic guitar
3. acrobatic gymnastics
4. acting in play
5. adjusting glasses
6. aerobics
7. air drumming
8. air travel
9. airbrush
10. alligator wrestling
11. alpine climbing
12. alpine skiing
13. amusement park
14. answering questions
15. applauding
16. applying cream
17. archaeological excavation
18. archery
19. arguing
20. arm wrestling
21. arranging flowers
22. arresting
23. assembling bicycle
24. assembling computer
25. attending conference
26. auctioning
27. baby transport
28. baby waking up
29. backflip (human)
30. backpacking (wilderness)
31. baking
32. baking cookies
33. balance beam
34. balloon blowing
35. bandaging
36. barbell
37. barbequing
38. bartending
39. base jumping
40. bathing
41. bathing dog
42. batting (cricket)
43. batting cage
44. battle rope training
45. beatboxing
46. bee keeping
47. belly dancing
48. bench pressing
49. bending back
50. bending metal
51. biceps curl
52. bicycling
53. biking through snow
54. blasting sand
55. blending fruit
56. blowing glass
57. blowing leaves
58. blowing nose
59. blowing out candles
60. bmx bike
61. boating
62. bobsledding
63. body piercing
64. bodyboarding
65. bodysurfing
66. bodyweight exercise
67. bookbinding
68. bottling
69. bouncing ball
70. bouncing ball (not juggling)
71. bouncing on bouncy castle
72. bouncing on trampoline
73. bowling
74. bowling (cricket)
75. braiding hair
76. breading or breadcrumbing
77. breakdancing
78. breaking

79. breaking boards
80. breaking glass
81. breathing fire
82. brush painting
83. brushing hair
84. brushing teeth
85. building cabinet
86. building lego
87. building sandcastle
88. building shed
89. bull fighting
90. bulldozer
91. bulldozing
92. bungee jumping
93. burping
94. busking
95. buttoning
96. cake decorating
97. calculating
98. calligraphy
99. camping
100. canoeing or kayaking
101. capoeira
102. caporales
103. capsizing
104. card stacking
105. card throwing
106. card tricks
107. carp fishing
108. carrying baby
109. carrying weight
110. cartwheeling
111. carving ice
112. carving marble
113. carving pumpkin
114. carving wood with a knife
115. casting fishing line
116. catching fish
117. catching or throwing baseball
118. catching or throwing frisbee
119. catching or throwing softball
120. celebrating
121. changing gear in car
122. changing oil
123. changing wheel
124. chasing
125. checking tires
126. checking watch
127. cheerleading
128. chiseling stone
129. chiseling wood
130. chopping meat
131. chopping vegetables
132. chopping wood
133. christmas
134. circus
135. clam digging
136. clay pottery making
137. clean and jerk
138. cleaning floor
139. cleaning gutters
140. cleaning pool
141. cleaning shoes
142. cleaning toilet
143. cleaning windows
144. climbing a rope
145. climbing ladder
146. climbing tree
147. closing door
148. coloring in
149. combat
150. comedian
151. concert
152. construction
153. contact juggling
154. contorting
155. cooking
156. cooking chicken
157. cooking egg
158. cooking on campfire
159. cooking sausages
160. cooking sausages (not on barbeque)
161. cooking scallops
162. cooking show
163. cosplaying
164. counting money
165. country line dancing
166. cracking knuckles
167. cracking neck

168. crawling baby
169. cricket
170. crocheting
171. crossing river
172. crouching
173. crying
174. cumbia
175. curling (sport)
176. curling hair
177. cutting apple
178. cutting cake
179. cutting nails
180. cutting orange
181. cutting pineapple
182. cutting watermelon
183. dancing
184. dancing ballet
185. dancing charleston
186. dancing gangnam style
187. dancing macarena
188. dashcam
189. deadlifting
190. dealing cards
191. decorating the christmas tree
192. decoupage
193. delivering mail
194. demolition
195. digging
196. dining
197. directing traffic
198. dirt track racing
199. disc golfing
200. disc jockey
201. diving cliff
202. docking boat
203. dodgeball
204. dog agility
205. doing aerobics
206. doing jigsaw puzzle
207. doing laundry
208. doing nails
209. doing sudoku
210. doing wheelie
211. drag racing
212. drawing
213. dressage
214. dribbling basketball
215. drifting (motorsport)
216. drinking
217. drinking beer
218. drinking shots
219. driving car
220. driving tractor
221. drooling
222. drop kicking
223. drumming fingers
224. dumbbell
225. dump truck
226. dumpster diving
227. dune buggy
228. dunking basketball
229. dying hair
230. eating burger
231. eating cake
232. eating carrots
233. eating chips
234. eating doughnuts
235. eating hotdog
236. eating ice cream
237. eating nachos
238. eating spaghetti
239. eating street food
240. eating watermelon
241. egg hunting
242. electric guitar
243. embroidering
244. embroidery
245. enduro
246. entering church
247. exercising arm
248. exercising with an exercise ball
249. explosion
250. extinguishing fire
251. extreme sport
252. faceplanting
253. falling off bike
254. falling off chair
255. feeding birds
256. feeding fish

257. feeding goats
258. building fence
259. fencing (sport)
260. festival
261. fidgeting
262. field hockey
263. figure skating
264. filling cake
265. filling eyebrows
266. finger snapping
267. fingerboard (skateboard)
268. firefighter
269. fireworks
270. fixing bicycle
271. fixing hair
272. flamenco
273. flint knapping
274. flipping bottle
275. flipping pancake
276. fly tying
277. flying kite
278. folding clothes
279. folding napkins
280. folding paper
281. forklift
282. french horn
283. front raises
284. frying
285. frying vegetables
286. gambling
287. garbage collecting
288. gardening
289. gargling
290. geocaching
291. getting a haircut
292. getting a piercing
293. getting a tattoo
294. giving or receiving award
295. gliding
296. go-kart
297. gold panning
298. golf chipping
299. golf driving
300. golf putting
301. gospel singing in church
302. greeting
303. grinding meat
304. grooming cat
305. grooming dog
306. grooming horse
307. gymnastics
308. gymnastics tumbling
309. hammer throw
310. hand washing clothes
311. head stand
312. headbanging
313. headbutting
314. heavy equipment
315. helmet diving
316. herding cattle
317. high fiving
318. high jump
319. high kick
320. hiking
321. historical reenactment
322. hitchhiking
323. hitting baseball
324. hockey stop
325. holding snake
326. home improvement
327. home roasting coffee
328. hopscotch
329. horse racing
330. hoverboarding
331. huddling
332. hugging
333. hugging (not baby)
334. hugging baby
335. hula hooping
336. hunting
337. hurdling
338. hurling (sport)
339. ice climbing
340. ice dancing
341. ice fishing
342. ice skating
343. ice swimming
344. inflating balloons
345. installing carpet

346. ironing
347. ironing hair
348. javelin throw
349. jaywalking
350. jetskiing
351. jogging
352. juggling
353. juggling balls
354. juggling fire
355. juggling soccer ball
356. jumping
357. jumping bicycle
358. jumping into pool
359. jumping jacks
360. jumping sofa
361. jumpstyle dancing
362. karaoke
363. kick (football)
364. kickboxing
365. kickflip
366. kicking field goal
367. kicking soccer ball
368. kissing
369. kitesurfing
370. knitting
371. krumping
372. land sailing
373. landing airplane
374. laughing
375. lawn mower racing
376. laying bricks
377. laying concrete
378. laying decking
379. laying stone
380. laying tiles
381. leatherworking
382. letting go of balloon
383. licking
384. lifting hat
385. lighting
386. lighting candle
387. lighting fire
388. listening with headphones
389. lock picking
390. logging
391. long jump
392. longboarding
393. looking at phone
394. looking in mirror
395. luge
396. lunge
397. making a cake
398. making a sandwich
399. making balloon shapes
400. making bed
401. making bubbles
402. making cheese
403. making horseshoes
404. making jewelry
405. making latte art
406. making paper aeroplanes
407. making pizza
408. making snowman
409. making sushi
410. making tea
411. making the bed
412. manicure
413. manufacturing
414. marching
415. marching band
416. marimba
417. marriage proposal
418. massaging back
419. massaging feet
420. massaging legs
421. massaging neck
422. mechanic
423. metal detecting
424. metal working
425. milking cow
426. milking goat
427. minibike
428. mixing colours
429. model building
430. monster truck
431. moon walking
432. mopping floor
433. mosh pit dancing
434. motocross

435. motorcycling
436. mountain biking
437. mountain climber (exercise)
438. moving baby
439. moving child
440. moving furniture
441. mowing lawn
442. mushroom foraging
443. musical ensemble
444. needle felting
445. news anchoring
446. news presenter
447. nightclub
448. none
449. offroading
450. ollie (skateboarding)
451. omelette
452. opening bottle
453. opening bottle (not wine)
454. opening coconuts
455. opening door
456. opening present
457. opening refrigerator
458. opening wine bottle
459. orchestra
460. origami
461. outdoor recreation
462. packing
463. parade
464. paragliding
465. parasailing
466. parkour
467. passing american football
468. passing soccer ball
469. peeling apples
470. peeling banana
471. peeling potatoes
472. penalty kick (association football)
473. person collecting garbage
474. personal computer
475. petting animal
476. petting animal (not cat)
477. petting cat
478. petting horse

479. photobombing
480. photocopying
481. picking apples
482. picking blueberries
483. picking fruit
484. pilates
485. pillow fight
486. pinching
487. pipe organ
488. pirouetting
489. planing wood
490. planting trees
491. plastering
492. playing accordion
493. playing american football
494. playing badminton
495. playing bagpipes
496. playing banjo
497. playing basketball
498. playing bass guitar
499. playing beer pong
500. playing billiards
501. playing blackjack
502. playing cards
503. playing cello
504. playing checkers
505. playing chess
506. playing clarinet
507. playing controller
508. playing cricket
509. playing cymbals
510. playing darts
511. playing didgeridoo
512. playing dominoes
513. playing drums
514. playing fiddle
515. playing field hockey
516. playing flute
517. playing gong
518. playing guitar
519. playing hand clapping games
520. playing handball
521. playing harmonica
522. playing harp
523. playing ice hockey

524. playing keyboard
525. playing kickball
526. playing laser tag
527. playing lute
528. playing mahjong
529. playing maracas
530. playing marbles
531. playing monopoly
532. playing netball
533. playing oboe
534. playing ocarina
535. playing organ
536. playing paintball
537. playing pan pipes
538. playing piano
539. playing piccolo
540. playing pinball
541. playing ping pong
542. playing poker
543. playing polo
544. playing recorder
545. playing road hockey
546. playing rounders
547. playing rubiks cube
548. playing rugby
549. playing saxophone
550. playing scrabble
551. playing shuffleboard
552. playing slot machine
553. playing snare drum
554. playing soccer
555. playing squash or racquetball
556. playing tennis
557. playing timbales
558. playing trombone
559. playing trumpet
560. playing tuba
561. playing ukulele
562. playing viola
563. playing violin
564. playing volleyball
565. playing with toys
566. playing with trains
567. playing xylophone
568. plumbing
569. poaching eggs
570. poking bellybutton
571. pole vault
572. polishing furniture
573. polishing metal
574. popping balloons
575. pouring beer
576. pouring milk
577. pouring wine
578. praying
579. preacher
580. preparing salad
581. presenting weather forecast
582. pretending to be a statue
583. protesting
584. pull ups
585. pulling
586. pulling espresso shot
587. pulling rope
588. pulling rope (game)
589. pumping fist
590. pumping gas
591. punching bag
592. punching person
593. push up
594. pushing car
595. pushing cart
596. pushing wheelbarrow
597. pushing wheelchair
598. putting on foundation
599. putting on lipstick
600. putting on sari
601. putting on shoes
602. putting wallpaper on wall
603. queuing
604. racing
605. radio-controlled model
606. rafting
607. rain
608. rallying
609. reading book
610. reading newspaper
611. recipe
612. recording music

613. recreational fishing
614. repairing puncture
615. riding a bike
616. riding camel
617. riding elephant
618. riding mechanical bull
619. riding mule
620. riding or walking with horse
621. riding scooter
622. riding snow blower
623. riding unicycle
624. ripping paper
625. roasting
626. roasting marshmallows
627. roasting pig
628. robot dancing
629. rock climbing
630. rock scissors paper
631. rocking
632. roller coaster
633. roller skating
634. rolling pastry
635. rope pushdown
636. rowing (sport)
637. running
638. running on treadmill
639. sailing
640. salsa dancing
641. saluting
642. sanding floor
643. sanding wood
644. sausage making
645. sawing wood
646. scrambling eggs
647. scrapbooking
648. screen printing
649. scrubbing face
650. scuba diving
651. seasoning food
652. separating eggs
653. serve (tennis)
654. setting table
655. sewing
656. shaking hands
657. shaking head
658. shaping bread dough
659. sharpening knives
660. sharpening pencil
661. shaving head
662. shaving legs
663. shearing sheep
664. shining flashlight
665. shining shoes
666. shooting basketball
667. shooting off fireworks
668. shopping
669. shot put
670. shouting
671. shoveling snow
672. shredding paper
673. shrugging
674. shucking oysters
675. shuffling cards
676. shuffling feet
677. side kick
678. sieving
679. sign language interpreting
680. silent disco
681. singing
682. sipping cup
683. situp
684. skateboarding
685. ski ballet
686. ski jumping
687. skiing crosscountry
688. skiing mono
689. skiing slalom
690. skipping rope
691. skipping stone
692. sky diving
693. skydiving
694. slacklining
695. slapping
696. sled dog racing
697. sleeping
698. slicing onion
699. slopestyle
700. smashing
701. smelling feet

702. smoking
703. smoking hookah
704. smoking pipe
705. smoothie
706. snatch weight lifting
707. sneezing
708. sniffing
709. snorkeling
710. snowboarding
711. snowkiting
712. snowmobile
713. snowmobiling
714. snowplow
715. snowshoe
716. soccer goal
717. somersaulting
718. sowing
719. speed skating
720. spelunking
721. spinning plates
722. spinning poi
723. splashing
724. splashing water
725. spray painting
726. spraying
727. springboard diving
728. square dancing
729. squat
730. squeezing orange
731. stacking cups
732. stacking dice
733. standing on hands
734. standup paddleboarding
735. staring
736. stealing
737. steer roping
738. steering car
739. sticking tongue out
740. stir frying
741. stirring
742. stomping grapes
743. street racing
744. stretching
745. stretching arm
746. stretching leg
747. strumming guitar
748. stunt performer
749. submerging
750. sucking lolly
751. sun tanning
752. surfing crowd
753. surfing water
754. surveying
755. sweeping floor
756. swimming
757. swimming backstroke
758. swimming breast stroke
759. swimming butterfly stroke
760. swimming front crawl
761. swimming with dolphins
762. swimming with sharks
763. swing dancing
764. swinging baseball bat
765. swinging legs
766. swinging on something
767. sword fighting
768. sword swallowing
769. tabla
770. tackling
771. tagging graffiti
772. tai chi
773. taking a shower
774. taking photo
775. talking on cell phone
776. tango dancing
777. tap dancing
778. tapping guitar
779. tapping pen
780. tasting beer
781. tasting food
782. tasting wine
783. teaching
784. tearing
785. telemark ski
786. tennis
787. testifying
788. texting
789. threading needle
790. throwing axe

791. throwing ball
792. throwing ball (not baseball or american football)
793. throwing discus
794. throwing knife
795. throwing snowballs
796. throwing tantrum
797. throwing water balloon
798. thunderstorm
799. tickling
800. tie dying
801. tightrope walking
802. tiptoeing
803. tobogganing
804. torte
805. tossing coin
806. tossing salad
807. train
808. training dog
809. trapezing
810. treating wood
811. trimming or shaving beard
812. trimming shrubs
813. trimming trees
814. triple jump
815. twiddling fingers
816. tying bow tie
817. tying knot (not on a tie)
818. tying necktie
819. tying shoe laces
820. tying tie
821. unboxing
822. uncorking champagne
823. underwater diving
824. unidentified flying object
825. unloading truck
826. using a microscope
827. using a paint roller
828. using a power drill
829. using a sledge hammer
830. using a wrench
831. using atm
832. using bagging machine
833. using circular saw
834. using computer
835. using inhaler
836. using megaphone
837. using puppets
838. using remote controller
839. using remote controller (not gaming)
840. using segway
841. vacuum cleaner
842. vacuuming car
843. vacuuming floor
844. valuting
845. visiting the zoo
846. volcano
847. wading through mud
848. wading through water
849. waiting in line
850. wakeboarding
851. waking up
852. walking on stilts
853. walking the dog
854. walking through snow
855. walking with crutches
856. washing
857. washing dishes
858. washing feet
859. washing hair
860. washing hands
861. washing machine
862. watching tv
863. water park
864. water skiing
865. water sliding
866. watercolor painting
867. waterfall
868. waterfowl hunting
869. watering plants
870. waving hand
871. waxing armpits
872. waxing back
873. waxing chest
874. waxing eyebrows
875. waxing legs
876. weaving basket
877. weaving fabric
878. wedding
879. weight lifting

```
action
├── activity
│       ├── applauding
│       ├── backflip
│       ├── backflip (human)
│       ├── bee keeping
│       ├── blowing nose
│       ├── blowing out candles
│       ├── bookbinding
│       ├── bottling
│       ├── braiding hair
│       ├── breaking
│       │       ├── breaking boards
│       │       └── breaking glass
│       ├── bungee jumping
│       ├── buttoning
│       ├── cartwheeling
│       ├── checking watch
│       ├── chewing
│       │       ├── blowing bubble gum
│       │       └── chewing gum
│       ├── clam digging
│       ├── clapping
│       ├── closing door
│       ├── cosplaying
│       ├── counting money
│       ├── crafting
│       │       ├── blowing glass
│       │       ├── carving
│       │       │       ├── carving ice
│       │       │       ├── carving marble
│       │       │       ├── carving pumpkin
│       │       │       └── carving wood with a knife
│       │       ├── chiseling stone
│       │       ├── chiseling wood
│       │       ├── clay pottery making
│       │       ├── crocheting
│       │       ├── decoupage
│       │       ├── folding paper
│       │       │       ├── making paper aeroplanes
│       │       │       └── origami
│       │       ├── knitting
│       │       ├── leatherworking
│       │       ├── model building
│       │       ├── ripping paper
│       │       ├── sewing
│       │       │       ├── embroidering
│       │       │       ├── embroidery
│       │       │       ├── needle felting
│       │       │       └── threading needle
│       │       ├── tearing
│       │       ├── weaving basket
│       │       ├── weaving fabric
│       │       ├── woodwoorking
```

880. welding
881. whistling
882. wildlife
883. windsurfing
884. winking
885. wood burning (art)
886. wood carving
887. woodworking

888. wrapping present
889. wrestling
890. writing
891. yarn spinning
892. yawning
893. yoga
894. zumba

# D  Full Hierarchy

```
action
├── activity
│       ├── applauding
│       ├── backflip
│       ├── backflip (human)
│       ├── bee keeping
│       ├── blowing nose
│       ├── blowing out candles
│       ├── bookbinding
│       ├── bottling
│       ├── braiding hair
│       ├── breaking
│       │       ├── breaking boards
│       │       └── breaking glass
│       ├── bungee jumping
│       ├── buttoning
│       ├── cartwheeling
│       ├── checking watch
│       ├── chewing
│       │       ├── blowing bubble gum
│       │       └── chewing gum
│       ├── clam digging
│       ├── clapping
│       ├── closing door
│       ├── cosplaying
│       ├── counting money
│       ├── crafting
│       │       ├── blowing glass
│       │       ├── carving
│       │       │       ├── carving ice
│       │       │       ├── carving marble
│       │       │       ├── carving pumpkin
│       │       │       └── carving wood with a knife
│       │       ├── chiseling stone
│       │       ├── chiseling wood
│       │       ├── clay pottery making
│       │       ├── crocheting
│       │       ├── decoupage
│       │       ├── folding paper
│       │       │       ├── making paper aeroplanes
│       │       │       └── origami
│       │       ├── knitting
│       │       ├── leatherworking
│       │       ├── model building
│       │       ├── ripping paper
│       │       ├── sewing
│       │       │       ├── embroidering
│       │       │       ├── embroidery
│       │       │       ├── needle felting
│       │       │       └── threading needle
│       │       ├── tearing
│       │       ├── weaving basket
│       │       ├── weaving fabric
│       │       ├── woodwoorking
```

```
action
├── activity
│       ├── applauding
│       ├── backflip
│       ├── backflip (human)
│       ├── bee keeping
│       ├── blowing nose
│       ├── blowing out candles
│       ├── bookbinding
│       ├── bottling
│       ├── braiding hair
│       ├── breaking
│       │       ├── breaking boards
│       │       └── breaking glass
│       ├── bungee jumping
│       ├── buttoning
│       ├── cartwheeling
│       ├── checking watch
│       ├── chewing
│       │       ├── blowing bubble gum
│       │       └── chewing gum
│       ├── clam digging
│       ├── clapping
│       ├── closing door
│       ├── cosplaying
│       ├── counting money
│       ├── crafting
│       │       ├── blowing glass
│       │       ├── carving
│       │       │       ├── carving ice
│       │       │       ├── carving marble
│       │       │       ├── carving pumpkin
│       │       │       └── carving wood with a knife
│       │       ├── chiseling stone
│       │       ├── chiseling wood
│       │       ├── clay pottery making
│       │       ├── crocheting
│       │       ├── decoupage
│       │       ├── folding paper
│       │       │       ├── making paper aeroplanes
│       │       │       └── origami
│       │       ├── knitting
│       │       ├── leatherworking
│       │       ├── model building
│       │       ├── ripping paper
│       │       ├── sewing
│       │       │       ├── embroidering
│       │       │       ├── embroidery
│       │       │       ├── needle felting
│       │       │       └── threading needle
│       │       ├── tearing
│       │       ├── weaving basket
│       │       ├── weaving fabric
│       │       ├── woodwoorking
```

```
action
├── activity
│       ├── applauding
│       ├── backflip
│       ├── backflip (human)
│       ├── bee keeping
│       ├── blowing nose
│       ├── blowing out candles
│       ├── bookbinding
│       ├── bottling
│       ├── braiding hair
│       ├── breaking
│       │       ├── breaking boards
│       │       └── breaking glass
│       ├── bungee jumping
│       ├── buttoning
│       ├── cartwheeling
│       ├── checking watch
│       ├── chewing
│       │       ├── blowing bubble gum
│       │       └── chewing gum
│       ├── clam digging
│       ├── clapping
│       ├── closing door
│       ├── cosplaying
│       ├── counting money
│       ├── crafting
│       │       ├── blowing glass
│       │       ├── carving
│       │       │       ├── carving ice
│       │       │       ├── carving marble
│       │       │       ├── carving pumpkin
│       │       │       └── carving wood with a knife
│       │       ├── chiseling stone
│       │       ├── chiseling wood
│       │       ├── clay pottery making
│       │       ├── crocheting
│       │       ├── decoupage
│       │       ├── folding paper
│       │       │       ├── making paper aeroplanes
│       │       │       └── origami
│       │       ├── knitting
│       │       ├── leatherworking
│       │       ├── model building
│       │       ├── ripping paper
│       │       ├── sewing
│       │       │       ├── embroidering
│       │       │       ├── embroidery
│       │       │       ├── needle felting
│       │       │       └── threading needle
│       │       ├── tearing
│       │       ├── weaving basket
│       │       ├── weaving fabric
│       │       ├── woodwoorking
```

```
action
└── activity
    ├── applauding
    ├── backflip
    ├── backflip (human)
    ├── bee keeping
    ├── blowing nose
    ├── blowing out candles
    ├── bookbinding
    ├── bottling
    ├── braiding hair
    ├── breaking
    │   ├── breaking boards
    │   └── breaking glass
    ├── bungee jumping
    ├── buttoning
    ├── cartwheeling
    ├── checking watch
    ├── chewing
    │   ├── blowing bubble gum
    │   └── chewing gum
    ├── clam digging
    ├── clapping
    ├── closing door
    ├── cosplaying
    ├── counting money
    ├── crafting
    │   ├── blowing glass
    │   ├── carving
    │   │   ├── carving ice
    │   │   ├── carving marble
    │   │   ├── carving pumpkin
    │   │   └── carving wood with a knife
    │   ├── chiseling stone
    │   ├── chiseling wood
    │   ├── clay pottery making
    │   ├── crocheting
    │   ├── decoupage
    │   ├── folding paper
    │   │   ├── making paper aeroplanes
    │   │   └── origami
    │   ├── knitting
    │   ├── leatherworking
    │   ├── model building
    │   ├── ripping paper
    │   ├── sewing
    │   │   ├── embroidering
    │   │   ├── embroidery
    │   │   ├── needle felting
    │   │   └── threading needle
    │   ├── tearing
    │   ├── weaving basket
    │   ├── weaving fabric
    │   └── woodwoorking
```

```
                    └── treating wood
            └── woodworking
                ├── planing wood
                ├── sanding wood
                ├── sawing wood
                ├── using circular saw
                ├── wood burning
                ├── wood burning (art)
                └── wood carving
├── crossing eyes
├── crouching
├── crying
│   └── throwing tantrum
├── dancing
│   ├── belly dancing
│   ├── breakdancing
│   ├── caporales
│   ├── country line dancing
│   ├── cumbia
│   ├── dancing ballet
│   ├── dancing charleston
│   ├── dancing gangnam style
│   ├── dancing macarena
│   ├── flamenco
│   ├── headbanging
│   ├── jumpstyle dancing
│   ├── krumping
│   ├── marimba
│   ├── moon walking
│   ├── mosh pit dancing
│   ├── nightclub
│   ├── pirouetting
│   ├── pumping fist
│   ├── robot dancing
│   ├── salsa dancing
│   ├── shuffling feet
│   ├── silent disco
│   ├── square dancing
│   ├── swing dancing
│   ├── tango dancing
│   └── tap dancing
├── drooling
├── dumpster diving
├── eating
│   ├── burping
│   ├── dining
│   │   └── setting table
│   ├── drinking
│   │   ├── drinking beer
│   │   ├── drinking shots
│   │   ├── sipping cup
│   │   ├── tasting beer
│   │   └── tasting wine
│   ├── eating burger
```

```
            ├── eating cake
            ├── eating carrots
            ├── eating chips
            ├── eating doughnuts
            ├── eating hotdog
            ├── eating ice cream
            ├── eating nachos
            ├── eating spaghetti
            ├── eating street food
            ├── eating watermelon
            ├── sucking lolly
            └── tasting food
    ├── entering church
    ├── exercise
            ├── aerobics
            ├── battle rope training
            ├── bodyweight exercise
            ├── canoeing or kayaking
            ├── doing aerobics
            ├── exercising arm
            ├── exercising with an exercise ball
            ├── gymnastics
                ├── acrobatic gymnastics
                ├── balance beam
                ├── gymnastics tumbling
                ├── somersaulting
                └── valuting
            ├── hula hooping
            ├── lunge
            ├── martial arts
                ├── capoeira
                └── kickboxing
            ├── mountain climber
            ├── mountain climber (exercise)
            ├── parkour
            ├── pilates
            ├── pull ups
            ├── punching
                ├── punching bag
                ├── punching person
                └── punching person (boxing)
            ├── push up
            ├── rope pushdown
            ├── running
                ├── chasing
                ├── jogging
                └── running on treadmill
            ├── situp
            ├── standing on hands
            ├── standup paddleboarding
            ├── stretching
                ├── bending back
                ├── contorting
                ├── cracking back
```

```
                            ├── cracking knuckles
                            ├── cracking neck
                            ├── stretching arm
                            ├── stretching leg
                            └── yoga
                    ├── tai chi
                    ├── walking
                    │   ├── crawling baby
                    │   ├── delivering mail
                    │   ├── jaywalking
                    │   ├── marching
                    │   ├── tightrope walking
                    │   ├── tiptoeing
                    │   ├── wading
                    │   │   ├── wading through mud
                    │   │   └── wading through water
                    │   ├── walking on stilts
                    │   ├── walking the dog
                    │   ├── walking through snow
                    │   └── walking with crutches
                    ├── weight lifting
                    │   ├── barbell
                    │   ├── bench pressing
                    │   ├── biceps curl
                    │   ├── carrying weight
                    │   ├── clean and jerk
                    │   ├── deadlifting
                    │   ├── dumbbell
                    │   ├── front raises
                    │   ├── snatch weight lifting
                    │   └── squat
                    └── zumba
        ├── falling
        │   ├── faceplanting
        │   ├── falling off bike
        │   └── falling off chair
        ├── fidgeting
        ├── finger movement
        │   ├── drumming fingers
        │   ├── finger snapping
        │   ├── fingerboard (skateboard)
        │   ├── tapping pen
        │   ├── twiddling fingers
        │   └── tying knot
        │       ├── tying bow tie
        │       ├── tying knot (not on a tie)
        │       ├── tying necktie
        │       ├── tying shoe laces
        │       └── tying tie
        ├── flipping bottle
        ├── flying kite
        ├── gambling
        │   ├── playing poker
        │   └── playing slot machine
```

```
├── garbage collecting
├── gliding
├── gold panning
├── head stand
├── historical reenactment
├── hitchhiking
├── jumping
│   ├── diving cliff
│   ├── jumping bicycle
│   ├── jumping into pool
│   ├── jumping jacks
│   ├── jumping sofa
│   ├── ski jumping
│   ├── skipping rope
│   └── triple jump
├── land sailing
├── laughing
├── letting go of balloon
├── licking
├── lifting hat
├── lighting candle
├── listening with headphones
├── lock picking
├── looking at phone
├── looking in mirror
├── making bubbles
├── making snowman
├── manipulating
│   ├── adjusting glasses
│   ├── arranging flowers
│   └── stacking
│       └── stacking cups
├── marriage proposal
├── metal detecting
├── moving
│   ├── carrying baby
│   ├── moving baby
│   ├── moving child
│   └── moving furniture
├── mushroom foraging
├── opening
│   ├── opening bottle
│   │   ├── opening bottle (not wine)
│   │   ├── opening wine bottle
│   │   └── uncorking champagne
│   ├── opening door
│   ├── opening present
│   ├── opening refrigerator
│   └── unboxing
├── paragliding
├── parasailing
├── person collecting garbage
├── pinching
├── playing
```

```
                  ├── bouncing ball
                  ├── bouncing ball (not juggling)
                  ├── bouncing on bouncy castle
                  ├── bouncing on trampoline
                  ├── building sandcastle
                  ├── egg hunting
                  ├── hopscotch
                  ├── playing american football
                  │   ├── kicking field goal
                  │   ├── passing american football
                  │   └── passing american football (in game)
                  ├── playing badminton
                  ├── playing board game
                  │   └── doing jigsaw puzzle
                  ├── playing controller
                  ├── playing games
                  │   ├── playing beer pong
                  │   ├── playing cards
                  │   │   ├── card stacking
                  │   │   ├── card throwing
                  │   │   ├── card tricks
                  │   │   ├── dealing cards
                  │   │   ├── playing blackjack
                  │   │   └── shuffling cards
                  │   ├── playing checkers
                  │   ├── playing chess
                  │   ├── playing dominoes
                  │   ├── playing mahjong
                  │   ├── playing monopoly
                  │   ├── playing pinball
                  │   ├── playing scrabble
                  │   ├── playing shuffleboard
                  │   └── rock scissors paper
                  ├── playing hand clapping games
                  ├── playing laser tag
                  ├── playing paintball
                  ├── playing toys
                  │   ├── playing marbles
                  │   ├── playing rubiks cube
                  │   └── radio-controlled model
                  ├── playing with toys
                  │   ├── building lego
                  │   ├── playing with trains
                  │   └── train
                  ├── pulling rope
                  ├── pulling rope (game)
                  ├── spinning plates
                  ├── spinning poi
                  ├── stacking dice
                  ├── using puppets
                  ├── using remote controller
                  ├── using remote controller (not gaming)
                  └── water sliding
          ├── praying
```

```
├── pretending to be a statue
├── pulling
├── pumping gas
├── pushing
│       ├── pushing car
│       ├── pushing cart
│       ├── pushing wheelbarrow
│       └── pushing wheelchair
├── reading
│       ├── reading book
│       └── reading newspaper
├── riding mechanical bull
├── rocking
├── saluting
├── scrapbooking
├── screen printing
├── shopping
├── shredding paper
├── shrugging
├── sieving
├── sign language interpreting
├── sitting
├── sky diving
├── skydiving
├── slacklining
├── sledding
│       └── tobogganing
├── smashing
├── smelling feet
├── smoking
│       ├── smoking hookah
│       └── smoking pipe
├── sneezing
├── sniffing
├── sports
│       ├── archery
│       ├── arm wrestling
│       ├── base jumping
│       ├── bicycle
│       │       └── assembling bicycle
│       ├── bobsledding
│       ├── bodyboarding
│       ├── bowling
│       ├── bull fighting
│       ├── catching or throwing frisbee
│       ├── catching or throwing softball
│       ├── cheerleading
│       ├── cricket
│       ├── curling
│       ├── curling (sport)
│       ├── diving
│       │       ├── springboard diving
│       │       └── underwater diving
│       ├── dodgeball
```

```
├── enduro
├── extreme sport
├── extreme sports
├── fencing
├── fencing (sport)
├── field hockey
├── frisbee
│   └── disc golfing
├── golfing
│   ├── golf chipping
│   ├── golf driving
│   └── golf putting
├── hammer throw
├── high jump
├── huddling
├── hurdling
├── hurling
├── hurling (sport)
├── ice skating
│   ├── figure skating
│   ├── ice dancing
│   └── speed skating
├── javelin throw
├── long jump
├── luge
├── playing baseball
│   ├── batting cage
│   ├── catching or throwing baseball
│   ├── hitting baseball
│   └── swinging baseball bat
├── playing basketball
│   ├── dribbling basketball
│   ├── dunking basketball
│   └── shooting basketball
├── playing billiards
├── playing cricket
│   ├── batting (cricket)
│   └── bowling (cricket)
├── playing darts
├── playing field hockey
├── playing handball
├── playing ice hockey
│   └── hockey stop
├── playing kickball
├── playing netball
├── playing ping pong
├── playing road hockey
├── playing rounders
├── playing rugby
├── playing soccer
│   ├── juggling soccer ball
│   ├── kick (football)
│   ├── kicking soccer ball
│   ├── passing soccer ball
```

```
                    ├── penalty kick (association football)
                    ├── shooting goal
                    ├── shooting goal (soccer)
                    └── soccer goal
            ├── playing squash or racquetball
            ├── playing tennis
            │   └── serve (tennis)
            ├── playing volleyball
            ├── pole vault
            ├── racing
            │   ├── dirt track racing
            │   ├── horse racing
            │   ├── lawn mower racing
            │   ├── racings
            │   └── sled dog racing
            ├── roller skating
            ├── rowing (sport)
            ├── shot put
            ├── skateboarding
            │   ├── kickflip
            │   ├── longboarding
            │   └── ollie (skateboarding)
            ├── snowboarding
            ├── snowkiting
            ├── surfing
            │   ├── bodysurfing
            │   ├── kitesurfing
            │   ├── surfing crowd
            │   ├── surfing water
            │   └── windsurfing
            └── tackling
├── spraying
├── sticking tongue out
├── stomping grapes
├── stunt performer
├── submerging
├── sun tanning
├── swinging
│   ├── swinging legs
│   └── swinging on something
├── taking photo
│   └── photobombing
├── talking
    ├── acting in play
    ├── answering questions
    ├── arguing
    ├── attending conference
    ├── auctioning
    ├── preacher
    ├── shouting
    ├── talking on cell phone
    ├── teaching
    ├── testifying
    └── using megaphone
```

```
├── throwing
│   ├── skipping stone
│   ├── throwing axe
│   ├── throwing ball
│   ├── throwing ball (not baseball or american football)
│   ├── throwing discus
│   ├── throwing knife
│   ├── throwing snowballs
│   └── throwing water balloon
├── tickling
├── tie dying
├── tossing coin
├── using phone
│   └── texting
├── waiting in line
├── whistling
├── winking
├── working
│   ├── surveying
│   └── unloading truck
├── writing
│   ├── calligraphy
│   └── doing sudoku
├── yarn spinning
└── yawning
animal
├── dog
│   └── dog agility
├── farming
│   ├── milking cow
│   ├── milking goat
│   └── steer roping
├── feeding birds
├── feeding fish
├── feeding goats
├── grooming
│   ├── grooming cat
│   ├── grooming dog
│   └── grooming horse
├── herding cattle
├── holding snake
├── riding camel
├── riding elephant
├── riding mule
├── riding or walking with horse
│   └── dressage
├── shearing sheep
├── training dog
├── visiting the zoo
└── wildlife
art
├── airbrush
├── drawing
│   └── coloring in
```

mixing colours
painting
brush painting
spray painting
tagging graffiti
using a paint roller
watercolor painting
cooking
baking
baking cookies
barbequing
cooking sausages
blending fruit
breading or breadcrumbing
cake decorating
cooking chicken
cooking egg
cooking on campfire
roasting marshmallows
cooking sausages (not on barbeque)
cooking scallops
cooking show
cutting
chopping meat
chopping vegetables
slicing onion
chopping wood
cutting cake
cutting fruit
cutting apple
cutting orange
cutting pineapple
cutting watermelon
opening coconuts
cutting nails
flipping pancake
frying
frying vegetables
grinding meat
home roasting coffee
making a cake
filling cake
torte
making a sandwich
making cheese
making latte art
making pizza
making sushi
making tea
omelette
peeling fruit
peeling apples
peeling banana
peeling potatoes

```
        ├── poaching eggs
        ├── pouring
        │       ├── pouring beer
        │       ├── pouring milk
        │       ├── pouring wine
        │       └── pulling espresso shot
        ├── preparing salad
        ├── recipe
        ├── roasting
        ├── roasting pig
        ├── rolling pastry
        ├── sausage making
        ├── scrambling eggs
        ├── seasoning food
        ├── separating eggs
        ├── shaping bread dough
        ├── sharpening knives
        ├── shucking oysters
        ├── smoothie
        ├── squeezing orange
        ├── stir frying
        ├── stirring
        └── tossing salad
├── entertainent
│   └── making balloon shapes
├── event
│   ├── celebrating
│   ├── circus
│   │   └── trapezing
│   ├── entertainment
│   │       ├── balloon blowing
│   │       ├── comedian
│   │       ├── inflating balloons
│   │       ├── juggling
│   │       │       ├── contact juggling
│   │       │       └── juggling balls
│   │       ├── popping balloons
│   │       └── sword swallowing
│   ├── festival
│   ├── giving or receiving award
│   ├── news
│   │       ├── news anchoring
│   │       ├── news presenter
│   │       └── presenting weather forecast
│   ├── parade
│   └── wedding
├── explosion
│   └── demolition
├── fire
│   ├── breathing fire
│   ├── extinguishing fire
│   ├── firefighter
│   ├── fireworks
│   ├── juggling fire
```

```
│       ├── lighting fire
│       └── shooting off fireworks
├── fun
│   ├── amusement park
│   ├── roller coaster
│   └── water park
├── interaction
│   ├── fighting
│   │   ├── combat
│   │   ├── headbutting
│   │   ├── high kick
│   │   ├── kicking
│   │   │   ├── drop kicking
│   │   │   └── side kick
│   │   ├── pillow fight
│   │   ├── slapping
│   │   ├── sword fighting
│   │   └── wrestling
│   ├── greeting
│   │   └── shaking hands
│   ├── high fiving
│   ├── kissing
│   ├── massage
│   │   ├── massaging back
│   │   ├── massaging feet
│   │   ├── massaging legs
│   │   ├── massaging neck
│   │   └── massaging person's head
│   ├── shaking head
│   ├── staring
│   ├── stealing
│   ├── touching
│   │   ├── hugging
│   │   │   ├── hugging (not baby)
│   │   │   └── hugging baby
│   │   └── poking bellybutton
├── music
│   ├── beatboxing
│   ├── concert
│   ├── disc jockey
│   ├── musical ensemble
│   ├── orchestra
│   ├── playing instrument
│   │   ├── acoustic guitar
│   │   ├── air drumming
│   │   ├── busking
│   │   ├── cymbal
│   │   ├── electric guitar
│   │   ├── fiddle
│   │   ├── french horn
│   │   ├── marching band
│   │   ├── pipe organ
│   │   ├── playing accordion
│   │   ├── playing bagpipes
```

```
                              ├── playing banjo
                              ├── playing bass guitar
                              ├── playing cello
                              ├── playing clarinet
                              ├── playing cymbals
                              ├── playing didgeridoo
                              ├── playing drums
                              │   └── tabla
                              ├── playing fiddle
                              ├── playing flute
                              ├── playing gong
                              ├── playing guitar
                              │   └── strumming guitar
                              ├── playing harmonica
                              ├── playing harp
                              ├── playing keyboard
                              ├── playing lute
                              ├── playing maracas
                              ├── playing oboe
                              ├── playing ocarina
                              ├── playing organ
                              ├── playing pan pipes
                              ├── playing piano
                              ├── playing piccolo
                              ├── playing recorder
                              ├── playing saxophone
                              ├── playing snare drum
                              ├── playing timbales
                              ├── playing trombone
                              ├── playing trumpet
                              ├── playing tuba
                              ├── playing ukulele
                              ├── playing viola
                              ├── playing violin
                              ├── playing xylophone
                              ├── snare drum
                              ├── tapping guitar
                              ├── timbales
                              └── viola
                      ├── recording music
                      └── singing
                              ├── gospel singing in church
                              └── karaoke
          ├── none
          └── outdoors
                  ├── abseiling
                  ├── alligator wrestling
                  ├── archaeological excavation
                  ├── backpacking (wilderness)
                  ├── camping
                  │   └── flint knapping
                  ├── climbing
                          ├── alpine climbing
                          ├── climbing a rope
```

```
│           │           ├── climbing ladder
│           │           ├── climbing tree
│           │           ├── ice climbing
│           │           └── rock climbing
│           ├── dashcam
│           ├── digging
│           ├── fishing
│           │   ├── carp fishing
│           │   ├── casting fishing line
│           │   ├── catching fish
│           │   ├── fly tying
│           │   ├── ice fishing
│           │   └── recreational fishing
│           ├── gardening
│           │   ├── picking fruit
│           │   │   ├── picking apples
│           │   │   └── picking blueberries
│           │   ├── planting trees
│           │   ├── sowing
│           │   ├── trimming shrubs
│           │   ├── trimming trees
│           │   └── watering plants
│           ├── hiking
│           │   ├── geocaching
│           │   ├── snowshoe
│           │   └── spelunking
│           ├── hunting
│           │   └── waterfowl hunting
│           ├── outdoor recreation
│           ├── riding snow blower
│           ├── skiing
│           │   ├── alpine skiing
│           │   ├── ski ballet
│           │   ├── skiing crosscountry
│           │   ├── skiing mono
│           │   ├── skiing slalom
│           │   ├── slopestyle
│           │   └── telemark ski
│           ├── volcano
│           ├── waterfall
│           └── yardwork
│               ├── blowing leaves
│               ├── mowing lawn
│               └── shoveling snow
├── personal computer
├── queuing
├── riding a bike
├── setting
│   ├── home activity
│   │   ├── baby waking up
│   │   ├── beauty
│   │   │   ├── blowdrying hair
│   │   │   ├── body piercing
│   │   │   ├── curling hair
```

```
            ├── doing nails
            ├── dyeing hair
            ├── dying hair
            ├── filling eyebrows
            ├── fixing hair
            ├── gargling
            ├── getting a haircut
            ├── getting a piercing
            ├── getting a tattoo
            ├── hair coloring
            ├── ironing hair
            ├── makeup
            │       ├── applying cream
            │       ├── putting on eyeliner
            │       ├── putting on foundation
            │       ├── putting on lipstick
            │       └── putting on mascara
            ├── manicure
            ├── scrubbing face
            ├── shaving
            │       ├── shaving head
            │       └── shaving legs
            ├── trimming or shaving beard
            └── waxing
                    ├── waving hand
                    ├── waxing armpits
                    ├── waxing back
                    ├── waxing chest
                    ├── waxing eyebrows
                    └── waxing legs
├── brushing
│       ├── brushing hair
│       └── brushing teeth
├── christmas
├── cleaning
        ├── bathing
        │       ├── bathing dog
        │       └── taking a shower
        ├── blasting sand
        ├── brushing floor
        ├── cleaning floor
        ├── cleaning gutters
        ├── cleaning pool
        ├── cleaning shoes
        ├── cleaning toilet
        ├── cleaning windows
        ├── making bed
        ├── making the bed
        ├── mopping floor
        ├── polishing furniture
        ├── shining shoes
        ├── sweeping floor
        └── vacuuming
                ├── vacuum cleaner
```

```
                        ├── vacuuming car
                        └── vacuuming floor
            ├── decorating the christmas tree
            ├── dressing
            │   ├── putting on sari
            │   └── putting on shoes
            ├── folding
            │   ├── folding clothes
            │   └── folding napkins
            ├── home improvement
            │   ├── installing carpet
            │   ├── laying tiles
            │   ├── plumbing
            │   ├── putting wallpaper on wall
            │   └── sanding floor
            ├── ironing
            ├── packing
            ├── sleeping
            ├── waking up
            ├── washing
            │   ├── washing dishes
            │   ├── washing feet
            │   ├── washing hair
            │   └── washing hands
            ├── washing clothes
            │   ├── doing laundry
            │   ├── hand washing clothes
            │   └── washing machine
            ├── watching tv
            └── wrapping present
    ├── lighting
    │   └── shining flashlight
    ├── medical
    │   ├── bandaging
    │   └── using inhaler
    └── office work
        ├── photocopying
        ├── sharpening pencil
        └── using computer
            └── using atm
├── snowmobile
├── tennis
├── toy
├── transportation
    ├── air travel
    │   ├── aircraft
    │   ├── landing airplane
    │   └── unidentified flying object
    ├── baby transport
    ├── bicycling
    │   ├── biking through snow
    │   ├── bmx bike
    │   ├── doing wheelie
    │   ├── minibike
```

```
        ├── building
        │   ├── building cabinet
        │   └── building shed
        ├── bulldozer
        ├── bulldozing
        ├── dump truck
        ├── heavy equipment
        ├── laying bricks
        ├── laying concrete
        ├── laying decking
        └── laying stone
├── electronics
│   └── assembling computer
├── forklift
├── logging
├── manufacturing
│   └── metal working
│       ├── bending metal
│       ├── making horseshoes
│       ├── making jewelry
│       ├── polishing metal
│       └── welding
├── mechanic
│   ├── changing oil
│   ├── changing wheel
│   ├── changing wheel (not on bike)
│   ├── checking tires
│   └── repairing puncture
├── plastering
├── police
│   ├── arresting
│   ├── directing traffic
│   └── protesting
└── using tools
    ├── using a microscope
    ├── using a power drill
    ├── using a sledge hammer
    ├── using a wrench
    └── using bagging machine
```

```
            ├── mountain biking
            └── riding unicycle
        ├── boating
        │   ├── capsizing
        │   ├── docking boat
        │   ├── jetskiing
        │   ├── sailing
        │   ├── wakeboarding
        │   └── water skiing
        ├── crossing river
        ├── driving
        │   ├── driving car
        │   │   ├── changing gear in car
        │   │   ├── drag racing
        │   │   ├── drifting (motorsport)
        │   │   ├── offroading
        │   │   │   ├── dune buggy
        │   │   │   └── snowmobiling
        │   │   ├── rallying
        │   │   ├── steering car
        │   │   └── street racing
        │   ├── driving tractor
        │   ├── go-kart
        │   ├── monster truck
        │   ├── motorcycling
        │   │   └── motocross
        │   └── snowplow
        ├── hoverboarding
        ├── riding scooter
        └── using segway
├── walking or riding with horse
│   └── playing polo
├── water
│   ├── rafting
│   ├── splashing
│   │   └── splashing water
│   └── swimming
│       ├── ice swimming
│       ├── swimming backstroke
│       ├── swimming breast stroke
│       ├── swimming butterfly stroke
│       ├── swimming front crawl
│       ├── swimming with dolphins
│       ├── swimming with sharks
│       └── underwater swimming
│           ├── helmet diving
│           ├── scuba diving
│           └── snorkeling
├── weather
│   ├── rain
│   └── thunderstorm
└── work activity
    ├── bartending
    ├── construction
```

Figure 1: Full AViD hierarchy