[Reviews · NeurIPS 2020]

Review 1

Summary and Contributions: This paper describes a new large-scale video dataset for action recognition. The dataset is designed to resolve the diversity, privacy, and expiration issues present in the existing datasets such as Kinetics [Kay 2017] or Moments in Time [Monfort 2018]. The paper describes the procedures to build the fully-annotated dataset and several studies to examine the statistics and baseline performance of the dataset.

Strengths: The proposed dataset is a clear contribution to the computer vision community; as described in Sec 1, existing datasets have been having various issues involving the diversity, privacy, and availability / license problems, and they are major reasons why there was no ImageNet-like common benchmark in video action recognition. This work potentially resolves a majority of these problems and serves as a common important resource for the future study in video recognition. As a dataset paper, this work presents several convincing studies to show that the proposed dataset resolves diversity and other issues compared to the previous work.

Weaknesses: I do not find a major issue in this paper. This is probably unavoidable but the long-tailed distribution (Fig 5 and 6) seems to pose a challenge for rare-category recognition. It would be good if there is any technical suggestion for less frequent labels.

Correctness: The dataset construction process and the empirical evaluation protocol look reasonable and appropriate.

Clarity: The paper is clearly written and easy to follow in overall. (Minor) There is one sentence I could not get: > (Sec 1) Experimentally, we show diversity and lack of diversity affects the recognition.

Relation to Prior Work: The paper shows sufficient empirical studies to contrast the previous datasets, and sufficiently discusses the motivational distinctions throughout the paper.

Reproducibility: Yes

Additional Feedback: This work really tackles on the difficult dataset issues in video recognition research. I really appreciate. UPDATE: Thanks for the author response, this work definitely makes a contribution. The final rating is the same.


Review 2

Summary and Contributions: The authors collect a video dataset AViD for action recognition by collecting videos from different countries, in contrast with previous datasets, which are mainly from North America. The faces in the videos are blurred, and they also make sure that the collected videos are licensed so that the dataset keep static. Update: The authors partly addressed my concerns, and I am raising my ratings from 4 to 5.

Strengths: This paper considers the data imbalance issue in terms of countries and cultures, which I am really glad to see. I think this is an important problem in fair AI. Secondly, it has been a long existing problem that original videos on the Internet (e.g. YouTube) that are included in CV datasets get deleted over time, and it is rather cumbersome for both authors and other researchers since usually the only option is to ask the original authors for raw videos, and so I am also glad to see this factor being considered during data collection.

Weaknesses: 1) The same action could mean very different, even opposite things in different cultures (such as nodding and shaking your head). The paper does not discuss how often this happens in the dataset, and how it effects performance of models trained or tested on this dataset. 2) The results in table 5 and 6 show that models trained on AViD surpass the ones trained on other datasets. However, it is unclear where the improvements come from (larger training size of AViD, country diversity, or better video quality). 3) I feel that the paper does not have sufficient evidence to support their claim that models trained on AViD are more diverse in terms of countries. Only table 4 is related to this topic. Since this is the paper's main contribution, I feel that the authors need more comprehensive experiments to support it. 4) From table 3, AViD is still quite unbalanced (even under the simplest metric). Being a benchmark dataset that claim to be from diverse countries, AViD might lead to other researchers claiming their models to be diverse and fair simply because they train their models on AViD, which is still inherently biased. I am not an expert on this, but I think there should be a discussion.

Correctness: Yes.

Clarity: The paper is not perfectly written, but I can understand most of it.

Relation to Prior Work: Yes.

Reproducibility: Yes

Additional Feedback: See the weaknesses section.


Review 3

Summary and Contributions: The authors propose a new video action recognition dataset, where the videos are made from diverse countries, static with creative license and blurred to protect the privacy. The authors demonstrate the necessity of creating such a dataset by comprehensive benchmark experiments. This work will be impactful for the community.

Strengths: 1. A new diverse, static, and privacy-protected video action recognition dataset. 2. It is large enough compared to existing datasets. 3. The authors demonstrate good generalization of the models pretrained on the proposed dataset. And sufficient detailed analysis is provided.

Weaknesses: 1. Currently, all the videos are short videos. However, it is unlikely that all the source videos are already well-trimmed. If they are not, then how are the temporal boundary are determined and will these annotations be released? I would like to invite the authors to provide more clarification on this. 2. Is there a human verification or voting mechanism to make sure human annotations accurate?

Correctness: Yes

Clarity: Yes.

Relation to Prior Work: Yes

Reproducibility: Yes

Additional Feedback: Final rating: The author's response partially address my concerns. However, I still feel the authors could also release the temporal annotation, which will be of great importance for understanding the temporal boundaries of those actions. In terms of concerns of R2, I agree they are reasonable. But I feel some of them are actually open research problems, e.g., how to make sure the actions have no ambiguity or how to male sure the videos collected from different countries are really diverse in terns of content. Therefore, I decide to maintain my rating.

[Author Response · NeurIPS 2020]

**R1:** Thank you for your review and understanding the potential impact of this dataset. We will clarify the sentence which is stating that we study the effect that diversity has on recognition and add discussion on the long-tailed distribution of actions and way for it to be addressed, such as weighting the loss per-class or sampling classes in the tail more frequently.

**R2:** Thank you for your review, you raised some good points we would like to further address.

1. Yes, we agree that different actions in different cultures have different meanings. This is one of the major motivations for creating the dataset. When selecting the action classes, we did our best to avoid situations where the action meaning could be ambiguous. For example, we would label an action as 'nodding head' rather than including the meaning (i.e., 'nodding yes'). We will add further discussion of this point and closely check all the classes to find any that may have ambiguity.

2. AViD is roughly the same size as Kinetics, HACS, etc., as we tried to illustrate in Table 1. Note that pre-training with AViD outperforms pre-training with Kinetics-600, despite that Kinetics-600 is a bit larger in terms of the video hours. We will add further experiments using subsets of AViD to more closely analyze how the size impacts performance. Similarly, we will add more experiments training on different diversity splits of AViD to determine how that compares, in the final version of the paper. The video quality (image resolution and frame rate) of AViD matches existing datasets like Kinetics.

3. We believe we provide statistical measurements showing that the AViD dataset sufficiently differs from the existing datasets in terms of the diversity. These include the heatmaps in Figure 1 as well as the data comparison in Table 3. In addition, we believe the experiments in Table 4 explicitly confirms the downside of the existing datasets compared to AViD. We will add more fine-grained experiments in the final version of the paper, such as training on one country/continent and testing on another.

4. We will add more discussion on the diversity in the paper. One discussion point is the population prior; the metric in Table 3 assumes a uniform distribution across countries, but this ignores aspects like population density. Places like the Sahara desert have a very low population but are weighted the same as places like India. We will add another metric comparing the distribution of videos to a distribution based on population density. We will further discuss these metrics and their drawbacks to more clearly state and explain the diversity of AViD and any potential limitations of it.

Overall, we believe the introduction of the AViD dataset will make a strong positive impact to the community, particularly compared to the standard practice today: using heavily biased video datasets (e.g., Kinetics with 90% of videos from North America) for both training and evaluation.

**R3:** Thank you for your review. To answer your questions:

1. All the videos are trimmed clips from longer ones. These intervals were annotated and checked by humans. We do not have the original untrimmed clips due to data storage limits ( 1TB for the full, untrimmed version) and anonymization difficulties. Using only the trimmed clips has been standard on many other datasets like Kinetics, moments in time, etc., and we tried to follow this while enhancing the country diversity, privacy, and stability in AViD.

2. Yes, during the annotation process, we did have attention checks of the annotators where they were asked to label videos we had manually done. If they failed those videos, their annotations were discarded. The annotators were asked to label 10-15 videos per task, and we further manually checked one video from each reviewer (in addition to the attention check) to ensure they were doing a good job. While there is some noise in the annotation process, overall we are confident the annotations are accurate. We will add this description to the paper.

[Meta-Review · NeurIPS 2020]

The paper describes a new large-scale dataset for action recognition, containing actions across many different countries, where faces are blurred for privacy reasons, and all videos have the Creative Commons license. R2 considers the dataset will be beneficial to the community, but recommends rejection, arguing that a more comprehensive experimental analysis related to bias would be needed for the paper to be accepted. While the concerns raised by R2 are legitimate, both the AC and SAC agree with R1 and R3 that the paper passes the acceptance bar of NeurIPS, as this work is a very important step towards datasets that are designed for everyone, for any culture, while attempting to protect privacy and copyright. The authors should add the clarifications in the rebuttal to the camera-ready version.